# Linear/Ladder-Like Polysiloxane Block Copolymers with Methyl-, Trifluoropropyl- and Phenyl- Siloxane Units for Surface Modification

**DOI:** 10.3390/polym13132063

**Published:** 2021-06-23

**Authors:** Stepan A. Ostanin, Alexei V. Kalinin, Yurij Yu. Bratsyhin, Natalia N. Saprykina, Vjacheslav V. Zuev

**Affiliations:** 1ITMO University, Kronverkskiy pr. 49, 197101 Saint-Petersburg, Russia; stepan.ostanin1995@gmail.com; 2S.V.Lebedev State Institute of Synthetic Rubber, 1, Gapsalskaya st. 1, 198035 Saint-Petersburg, Russia; ak74frog@mail.ru (A.V.K.); trum@mail.ru (Y.Y.B.); 3Institute of Macromolecular Compounds of the Russian Academy of Sciences, Bolshoi pr. 31, 199004 Saint-Petersburg, Russia; elmic@hq.macro.ru

**Keywords:** block copolymers, ladder-like structure, polydimethylsiloxane, morphology, adhesion

## Abstract

Multiblock copolymers containing linear polydimethylsiloxane or polymethyltrifluoropropylsiloxane and ladder-like polyphenylsiloxane were synthesized in a one-step pathway. The functional homopolymer blocks and final multiblock copolymers were characterized using solution and solid-state multinuclear ^1^H, ^13^C, ^19^F, and ^29^Si NMR spectroscopy. It was shown that the ladder-like block contains silanol units, which influence the adhesion properties of multiblock copolymers and morphology of their casted films. The adhesion to metals and mechanical properties of multiblock copolymers were tested. The SEM study of casted films of multiblock copolymers shows the variety of formed morphologies, including long-strip-like or globular.

## 1. Introduction

The development of new coatings for advanced applications requires materials with the internal capability for self-organization on a micro-nano level. One of many pathways to solve this problem is the utilization of block-copolymers with complex architecture. A combination of blocks with different structure, length, and chemical compositions, with a variation of their sequence and consistency, allows to create materials with enormous versatility for application [1]. Because of the dual nature of their backbones (organic/inorganic), silicone containing blocks are widely used in the preparation of block-copolymers [2]. This topic is directly related to the interesting combination of properties offered by these fragments, which include extremely high backbone flexibility, good thermal and oxidative stability, high gas permeability, excellent dielectric properties, and physiological inertness or biocompatibility [3].

Polydimethylsiloxanes (PDMS) and their trifluoropropyl substituted analogues display very low surface tension values [4], providing an opportunity for the development of materials with precisely structured elements in two and three dimensions on length scales from many microns down to several nanometers [5].

The polyhedral oligomeric silsesquioxanes (POSS) with their rigid high thermostable structures present another structural motive for the variation of architecture of block-copolymers [6]. The major usage of such polymers is in coating/thin film applications [7]. The advantage of POSS in relation to the random crosslinked linear polysiloxanes is that they remain processable by dissolution or softening/melting. The most popular pathway for POSS synthesis throughout the polycondensation of trichlorophenylsilanes leads to the formation of an irregular body with ladder and different cage structures [8]. However, such ladder-like polysiloxanes show much higher thermal stability, radiation resistance, and mechanical strength compared to their single chain and branched species [6]. The preparation of ladder-like polysiloxanes is usually cost efficient and the presence of end and inner hydroxyl groups allows using a simple chemistry to synthesize block copolymers with very variable structures [9].

This paper focuses on synthesis of PDMS/ladder-like and its trifluoropropyl substituted analogues block-copolymers. The main properties of such polymers are significantly altered by various critical structural components. Many studies have been dedicated to understanding their structures. However, controversial interpretations of the analytical results with regard to the siloxane structure, such as ladder- and cage-like or random, have often been reported [10]. It is the objective of this work to offer insights into the structure of synthesized block-polymers. Another aim of this work is to study the self-assembly of the synthesized block-copolymers.

## 2. Experimental Part

### 2.1. Materials

The solvents (toluene, butyl acetate, chloroform, xylene (mixture of isomers), benzene, and tetrahydrofuran (THF)) were purchased from Reachim (Saint Petersburg, Russia), then distillated and purified by standard methods before use. N,N-diethylhydroxylamine (DEHA) was purchased from Shanghai Sunwise Chemical Co., Ltd. (PRC) (Shanghai, China) and used as received unless otherwise stated. Silanol-terminated polydimethylsiloxane (PDMS, I, m = 111, M_n_ = 8400), polymethyltrifluoropropylsiloxane (PMFS, II, m = 101, M_n_ = 16,000), and ladder-like polyphenylsiloxane (LAD, III, m = 15, M_n_ = 3950) were purchased from FGUP NIISK (Saint Petersburg, Russia). Vinyltris(methylethylketoximino)silane (VOS) was supplied by Mitsibishi Chemical Ltd. (Tokyo, Japan).

### 2.2. Synthetic Procedure

The block-copolymers L-PDMS IV and L-PMFS V were synthesized in a single-step reaction, as shown in Scheme 1 and described below.

Briefly, 15 g of LAD III was dissolved in 120 mL of toluene in a three-neck round-bottom flask with condenser and stirrer. Then, 15 g of PDMS I was added as solution in 120 mL of butyl acetate, and the mixture was heated to 110 °C. At this temperature, 0.2 g of DEHA was added and the solution was stirred for 2 h. After that, the solvents were removed by steam distillation and block-copolymer IV was dried at 120–130 °C in vacuum. The yield is practically quantitative.

Block-copolymer L-PMFS V was synthesized by the same method with change of PDMS I for PMFS II.

### 2.3. Analytical Methods

#### 2.3.1. Contact Angle Measurements

Drop Shape Analyzer DSA100E KRUSS (Hamburg, Germany) was employed to study the contact angle of a water droplet on the block-copolymer films by the sessile drop method. An average of 5 measurements taken on polymer samples coated on 5 separate metal substrates is reported as the water contact angle.

#### 2.3.2. Liquid Chromatography

Gel permeation chromatography was conducted on a Breeze QS HPLC (Waters Corporation, Milford, MA, USA) equipped with three different columns connected in series (HR-1, HR-2 and HR-4). Two detectors were employed in this study, a UV (Ultraviolet) Waters 2487 operated at a wavelength of 254 nm and differential refractive index Waters 2414. Tetrahydrofuran (THF) was used as eluent at a flow rate of 0.35 mL/min at 35 °C. The system was calibrated using polystyrene standards with different Mw obtained from Sigma-Aldrich (St. Louis, MO, USA) using the universal calibration method.

#### 2.3.3. FTIR Spectroscopy

Attenuated total reflectance infrared spectroscopy (ATR-FTIR) was conducted on a Vertex 50 FTIR (Bruker, Billerica, MA, USA) with a universal attenuated total reflection (ATR) sampling accessory on a diamond crystal. Spectra of the different products from the synthesis were taken with 2 cm^−1^ resolution and 32 scans in the range of 4000–650 cm^−1^ at 25 °C.

#### 2.3.4. NMR Spectroscopy

^1^H, ^13^C, ^19^F, and ^29^Si solution NMR spectra were recorded on an AVANCE 400 (Bruker, Billerica, MA, USA) spectrometer. ^1^H, ^13^C, and ^19^F NMR spectra were recorded in CDCl_3_ solution. For recording ^29^Si solution NMR spectra, a home-made Teflon tube was used. Spectra were recorded in THF solution.

^1^H NMR spectra were recorded on an AVANCE II 500 (Bruker, Billerica, MA, USA) spectrometer. Solid-state NMR experiments were performed on a Bruker Avance II 500 spectrometer operating at a proton frequency of 500.13 MHz. Further, 4 mm and 7 mm double resonance MAS probe heads were used for recording ^13^C and ^29^Si spectra under MAS conditions and for ^1^H, ^13^C, and ^29^Si static experiments correspondingly. For recording ^13^C and ^29^Si spectra of block-copolymers V under MAS conditions, a mixture of 1:1 with Al_2_O_3_ powder was used. Conventional cross-polarization MAS NMR experiment (CP-MAS) with high power proton decoupling (100 kHz) was used to detect the signals from rigid component with high CP efficiency. The recycle delay was 5 s and CP contact time 500 µs. ^13^C direct-polarization MAS NMR experiment with echo detection (90°-τ-180°-τ-aq) and low power proton decoupling (LPD, 25 kHz) was used to observe the mobile component. Applying 180° pulse after τ delay refocuses the mobile component, whereas the broad signals from the rigid part are already irreversibly relaxed [11]. The recycle delay in this experiment was 0.5 s (optimal for short relaxing of mobile component), and the echo delay τ 3 ms. Spinning frequency in all experiments was 5–8 kHz. Spectra were recorded at ambient temperature.

#### 2.3.5. Thermal Analysis

The thermal decomposition behavior of the block-polymers was studied using TA instruments TG 209 F3 Tarsus (Netzsch, Selb, Germany). About 10 mg of sample was taken in an alumina crucible and the thermogravimetric analysis was carried out from room temperature to 800 °C at a heating rate of 10 °C/min under nitrogen (99.999% pure) atmosphere at a flow rate of 100 mL/min. The glass transition temperature (T_g_) was measured using DSC instrument 204 F1 Phoenix (Netzsch, Selb, Germany). The analysis was done at heating rate of 10 °C/min in the temperature range of −150 to 200 °C.

#### 2.3.6. Mechanical Properties

Mechanical properties of the copolymer IV, such as tensile strength and elongation at break, were measured using a Flexural Test Setup Shimadzu Universal Testing Machine (UTM) as per ASTM D412 standard. For this, dumb-bell shaped specimens conforming to Type II of ASTM standard were cut from casting films of the copolymers using a die. For mechanical tests of copolymer V, their films were cut using stainless steel scissors into square samples (20 × 20 × 2 mm). The samples were subjected to mechanical puncturing using a texture analyzer TA.XTplus (Stable Micro Systems, Surrey, UK). The sample was fixed in a horizontal position with the film support rig HDP/FSR installed on a metal platform HDP/90, creating a 60-mm gap between the lower working surface of the texture analyzer and the film support rig for the passage of the puncturing tip through the sample. A load cell (30 kg) of the texture analyzer measured the resistance load of the sample during puncture with a steel spherical probe P/5S (Ø = 5 mm) at a speed of 1 mm/s using a special Exponent software. During the test, the dependence of the applied force on the distance for the deepening of the probe piercing the film was determined. An average of 5 measurements is reported.

#### 2.3.7. Adhesion Strength Testing

The adhesion of copolymers to metals was tested according to ISO 4624:2002 by the destructive testing of the bonded coupons using PosiTest AT-M Manual Adhesion Tester (DeFelsko, Ogdensburg, NY, USA). An average of 5 measurements is reported.

#### 2.3.8. Microscopy

A Supra 55 VP scanning electron microscope (SEM) (Carl Zeiss, Jena, Germany) was used to analyze the morphology of the block-copolymers. 

## 3. Results and Discussion

### 3.1. Synthesis of Block-Copolymers

The commercially available (α,ω) hydroxyl terminated silicone oligomers free of ethoxylated end groups were chosen for this study. PDMS and more pronounced polytrifluoropropylmetylsiloxane (PTFPS) have poor mechanical properties, making their reinforcement essential. LAD III having remaining reactive silanol end and inner groups with their ladder-like structure seem the best additive for the improvement of mechanical properties of siloxanes [12]. The molecular structure of POSS largely depends on the synthetic methods and processing, which can be classified into randomly branched, ladder-like, cage and partial cage structures. These structural variations govern the properties of the resulting block-copolymers. The pathway of synthesis of block-copolymers IV and V is presented on Scheme 1. Polyfunctionality of LAD III leads to the formation of a complex branched structure of block-copolymers IV and V. Scheme 1 presents only some examples from a numbers of possible variants. However, the selected molar ratio between LAD III and PDMS/PTFPS oligomers, which is much lower than equivalent prevents the formation of crosslinked structures and leads to formation of soluble products. The use at synthesis of the same weight of PDMS and PTFPS, despite the large difference in the mass of monomer units, has been determined the difference in molar mass of the prepared block-polymers because bigger difference in the number of hydroxyl groups in LAD III and PTFPS oligomer. The molar mass of the prepared block-polymers was determined with liquid chromatography (for IV M_w_ = 9.0 × 10^4^, M_w_/M_n_ = 1.97; for V M_w_ = 4.2 × 10^4^, M_w_/M_n_ = 2.32). The obtained values support the formation of multiblock-copolymers composed from each of three or four initially used oligomeric units. The formation of multiblock-copolymers, but not a mixture of starting components is supported by a relatively narrow product peak in chromatograms. The formation of a mixture should lead to the observation of a broad signal with the presence of initially low molecular components. The narrow molecular mass distributions of block-copolymers IV and V also speak to the higher reactivity of silanol groups under the chosen conditions.

The structure of multiblock-copolymers was well confirmed by IR spectroscopy. As one can see in Figure 1, the spectra of block-copolymers IV and V appear as superpositions of IR spectra of starting blocks. 

Only in the spectrum of oligomeric III can one observe the absorption of silanol OH-groups at 3500–3200 cm^−1^ (free and H-bonded groups) [13]. The IR spectra contain the vibrations of C–H, C–C, Si–O, Si–C, and C–F bonds in accordance with common spectroscopic analysis [13].

The solution ^1^H NMR spectra of block copolymers IV and V (Figure 2) confirm the structure of copolymers.

Therefore, the presence of OH groups signals relates to the defected ladder-like structure of the ladder block (the presence of silanol units). In block-copolymers, this signal is located around 1.6 ppm. Therefore, this signal is common with water presented in deuterochloroform (1.56 ppm). To support the presence of silanol OH groups in block-copolymers, titration was performed using of Grignard reagents. The presence of silanol units was confirmed in both block-copolymers. The fraction of OH groups in block-copolymer L-PMFS V is about four-fold higher than in L-PDMS IV, in accordance with the mole ration of LAD and PDMS/PMFS at synthesis.

A similar situation is observed in the ^1^H NMR spectrum of LAD III. The downfield shift of the OH-groups signal is connected with the shielding effect.

The presence of defects in this block was confirmed by solution ^13^C NMR spectra (Figure 3).

The redundant and broad signals of ipso carbon of phenyl groups indicates the presence of defects in the ladder structure. This can be confirmed by ^29^Si NMR spectroscopy. The solid state NMR spectra are given in Figure 4.

As one can see, these spectra contain not only signals from T_3_ units (centered at −82 ppm) that correspond to a perfect ladder structure, but also minor signal from −69 to −72 ppm related to T_2_ units of cage-like structure or silanol fragments [14]. Hence, we certify the structure of block-copolymers IV and V (the chemical shift of fluorine in trifluoromethyl group of V is −65.5 ppm, see Appendix A). The most important result of this study is the fact that the ladder-like block of copolymers contains defects with silanol groups that can result in a decrease of the contact angle, deterioration of mechanical properties, and to an increase of adhesion. 

Another important result obtained in the NMR study entails data concerning molecular dynamics. As one can see from solution ^1^H NMR spectra of block copolymers IV and V (Figure 2), the signals of aromatic protons are very broad. This is connected to the restricted molecular mobility in ladder-like fragments. This is consistent with the solution ^13^C NMR spectra, where broad signals of aromatic units coexist with sharp signals of quartet of CF_3_ group, which can freely rotate without restriction (Appendix A). The distribution of rigid and flexible fragments of block-copolymers can be obtained from solid state ^1^H NMR spectra (Figure 5). 

As one can see, the spectra of block-copolymers IV and V contain both rigid (broad) and mobile (sharp) parts. The mobile part contains the protons with different resonances which are related not only to the different fragments, but to different mobile phases as well. The exact assignment of these resonances is very complex. The ^13^C CP/MAS spectra of polymers III–V (Figure 6) indicating fragments with restricted mobility contain the signals from both blocks used at synthesis. Hence, the formal rigid regions contain both ladder and linear units.

The solid state ^1^H NMR spectrum of oligomer III contains mostly rigid components (Figure 7). However, the presence of mobile components allows us to calculate the fraction of defects (silanols) in these oligomers, which is less than 10%.

Hence, we synthesized multiblock-copolymers with a complex rigid–soft structure with defect units containing silanol groups.

### 3.2. Thermal Properties

We performed a DSC analysis of block-copolymers IV and V (Figure 8). 

As one can see form DSC curves of block-copolymer IV, it behaves as pure PDMS with T_g_ about −124 °C, recrystallization at −92.5 °C, and melting at −46 °C [15]. Hence, at such lengths of PDMS blocks, the influence of ladder-like units has not been observed. Similar results was obtained for polymer V with T_g_ about −67.5 °C.

TG data (Figure 9) for both copolymers IV and V are very similar, and as one can see, the presence of ladder units did not affect thermostability in comparison with pure PDMS or PTFPS [15].

### 3.3. The Surface Properties of the Films of Copolymers IV-V

For the investigation of water contact angle of copolymers IV and V, their solutions in THF were casted on metal plate (steel, copper, aluminum) and annealed at 70 °C. Because the annealing temperature was much higher than the glass transition temperatures of copolymers, the casting solvent did not play any role in the surface formation of casting films. For copolymer IV, the measured water contact angle of film did not depend on water and is 107.5°. For copolymer V the following dependence is observed: water contact angle is 110.5° on aluminum, 111.3° on steel, and 115.7° on copper. However, SEM images of surfaces did not allow to find any differences (Figure 10a,b).

We measured an adhesion to these casting films. No obvious connection with the values of water contact angle of films was observed. The adhesion of copolymer IV to copper is 0.71 MPa, 0.96 MPa to aluminum, and 1.70 MPa to steel (with deviation between five samples less than 0.05 MPa). A similar range was obtained for copolymer V: 0.41, 0.46, and 0.60 MPa, correspondingly. Hence, obviously, low surface energy leads to poor adhesion. Adhesion is also connected with the mechanical properties of block-copolymers. Copolymers IV have good mechanical performance as compared to polysiloxanes [16]. Their tensile strength is 1.3 ± 0.1 MPa at elongation 370 ± 20%. Clearly, this result is connected with the presence of ladder units, which play the role of nanofibers. The mechanical performance of copolymer V is far worse. It is impossible to perform a test using dumb-bell shaped specimens on a breaking machine. Accordingly, we opted for mechanical puncturing using a texture analyzer. The tensile strength of copolymer V is 78 ± 5 kPa at elongation 150 ± 10%.

### 3.4. Solubility and Morphology of Block-Copolymers

After synthesis, copolymer IV is a white-yellowish powder and copolymer V is white rubber. As one can see from Table 1, the copolymer IV is well soluble in most polar solvents. The behavior of copolymer V is more complex. It can form emulsions in many solvents (Figure 11). 

There is a possibility to manipulate the morphology of casted films using different solvents and solution states.

The annealed films of both copolymers form the smooth films on the surface of support for recording SEM images distinctly replicating the dashes (Figure 10a,b). The difference in appearance of these two films casted from toluene is caused by the quality of this solvent for each copolymer. The toluene is a very good solvent for copolymer IV and, as a result, the film casted from it does not show any signs of phase separation or structuring. The opposite situation is observed for copolymer V. The poor solvent allows the process of phase separation of different blocks and their aggregation to occur (Figure 10b). 

The formation of block copolymer nanostructures is an intriguing phenomenon. However, these processes have been impaired by annealing at the drying stage of casting films. One of the most frequently employed strategies to overcome it is crosslinking at casting from solution [16]. We used such a popular silicone crosslinker as VOS. It was added to polymer solution in the amount of 1 wt.% to polymer just before casting. 

We start this pathway from copolymer IV. The good solubility of this polymer in many solvents limits the possibility for the setup of self-organization. However, we prepared the films from solution in chloroform and recorded their images from the surface and the chip (Figure 10c,d). As one can see, the surface is smooth, but the bulk structure is complex and consists of grains and layers. If the casted copolymer V film was crosslinked from solution in good solvent with respect to one block (butyl acetate), a similar picture (Figure 10e,f) was observed. Thereby, it was shown that the use of selectivity of solvent to any one block of copolymers opens a possibility to manipulate the self-organization of block copolymers. In the case when a solvent is good for both blocks, the situation is much simpler. When the film was crosslinked from such a solvent as hexafluorobenzene, the self-organization was not fixed (Figure 10g,h), and both the surface and chip of films are smooth. 

The formation of emulsion (or micellar solution, any kind of latex) presents a new possibility for the manipulation of self-organization. First of all, the crosslinking allows to fix a globular structure which is formed in solution (Figure 10i). Interestingly, at slow drying, such a structure can be transformed in a regular long-strip-like surface (Figure 10j,k). Similar structures have been formed with the preparation of films from latexes of polysiloxane/poly (fluorinated acrylate) block-copolymers [17]. The authors connected the formation of such structures to the core/shell segregation of block copolymers in solution. Hence, in our case, the observed film structures also confirm the segregation of blocks in solution. Thereby, the possibility for the fabrication of well-ordered block copolymer thin films by the variation of solvents used at film casting was demonstrated.

## 4. Conclusions

Linear/ladder-like polysiloxane block copolymers were synthesized by one pot synthesis using starting linear oligomeric polydimetylsiloxane or methyl(trifluoropropyl)siloxane and ladder-like phenylsiloxane. By using the combination of solution and solid state multinuclear NMR spectroscopy, it was shown that the ladder block contains defect silanol units that decrease the rigidity of the ladder block but increase the overall hydrophilicity of copolymers. The surface properties of block copolymers were studied using contact angle measurements. The presence of trifluoropropyl groups leads to the increase of the contact angle of casted films on metal (steel, copper, aluminum) plate. The copolymers are soluble in many organic solvents. However, the different solubility of constituent blocks in various solvents opens the possibility to manipulate the morphology of the casted films. It was shown that the combination of a variation of starting solvent with the use of crosslinking in solution allows to obtain the films of copolymers with globular or long-strip-like morphology. This open the possibility to use the synthesized block copolymers in lithographic applications.

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
