# Peer review of "Linear/Ladder-Like Polysiloxane Block Copolymers with Methyl-, Trifluoropropyl- and Phenyl- Siloxane Units for Surface Modification"

_polymers, 2021, doi:10.3390/polym13132063_

Round 1
Reviewer 1 Report
This paper reports the synthesis of polysiloxane multiblock copolymers containing linear and ladder-like blocks via condensation of end silanol groups. The obtained block copolymers were characterized by different techniques such as NMR, FT-IR and GPC. The application of these polymers for hydrophobic modification of metal surfaces was demonstrated. This is an interesting approach, since linear silicones are known for their low mechanical strength and the authors employed ladder-like POSS units to reinforce them. Generally speaking, the research is new, interesting and original; therefore, I would recommend accepting this manuscript after considering the following points.
(1) I am not totally convinced that block copolymers were obtained in this work. The reactivity of silanol groups in the phenyl-POSS is quite low, and no characterization methods showed unambiguous evidence of the formation of block copolymers. I know it is quite different, nevertheless, I would propose the authors to make affords to do it. For example, they may compare the GPC, NMR and thermal analysis data of their products with the reaction mixtures. The change of appearance of the mixture after the reaction as well as the solubility in certain organic solvents can also serve as a good proof of the copolymer formation.
(2) Figure 10 shows that the morphology of thin films depends on the solvent, from which they were cast. This might be an evidence for the mixture of polymers rather than block copolymers. Can the thin film morphology of the same polymer be unified after thermal annealing? By the way, in Fig. 10 the authors should indicate the solvent for each micrograph.
(3) There are some typos and errors in the text, just name a few.
Line 50, PDMS instead of PMDS.
Line 61, (THF)).
Line 92, wavelength instead of wavelengths
Line 172, GPC does not allow measuring exact molar mass. The word “exact” should be removed.
Author Response
Answer to comments reviewer 1.
- About formation of block –copolymers.
I am agree with reviewer that spectroscopic or TG data did not given one hundred percent confirmation of block-copolymer formation.
However, GPC data given this. As one can seen from GPC data resulting copolymers given on chromatograms one (one) relative narrow peak with molecular mass distribution lower than three b and middle mass about 40000-70000. In the case of mixture of starting oligomers or their admixing we should have on chromatograms the peaks or shoulders at lower MM. (Starting POSS have MM lower than 4000). In the case of formation block copolymers too we should have very broad peaks. It is not observed that reliable support the formation the block-copolymers.
- According Fig.10
The morphology of not cross-linked films (Fig.10a,b) support the formation of block-copolymers. As wrote (line 297-298) this is images of annelid films, casted from toluene. In the case mixture of oligomers annealing should leads to formation phase separated structure but not to smooth surface.
- Other suggestion will accepted.

Reviewer 2 Report
The manuscript titled "Linear/ ladder-like polysiloxane block copolymers with me-2 thyl-, trifluoropropyl- and phenyl- siloxane units for surface 3 modification" needs revision.
- The whole abstract needs to be rewritten. The significance and purpose of this research should be clearly presented in the abstract. The abstract must be presented in a clear way in problematic, objective, idea, description of idea, highlighting the methods, results, quantitative comparison of results with significant findings, conclusions.
- The introduction must be revised. Add more state-of-the-art comparisons for the proposed work. Then do a critical analysis of previous research. State explicitly the shortcomings of previous research. What is positive in previous research and what is negative. Based on that, you explicitly define the goal of the research and the scientific hypothesis.
- Highlight the novelty of your methodology.
- The biggest shortcoming of the research is that there is a poor presentation of Figures. Add intensity on the Y-axis to understand the observations.
- Add more details about the Adhesion strength testing.
- Highlight the observation in SEM images by arrows and circles.
- How did you choose the experiment setting? Elaborate experiments parameters.
- The Conclusion section should be rewritten. Highlight your scientific contribution. Highlight the benefits of your research. Define shortcomings and future research.
Author Response
Answer to comments reviewer 2.
- The whole abstract needs to be rewritten. The significance and purpose of this research should be clearly presented in the abstract. The abstract must be presented in a clear way in problematic, objective, idea, description of idea, highlighting the methods, results, quantitative comparison of results with significant findings, conclusions.
I think that exist any differences in scientific culture. The suggestion of reviewer is acceptable for Introduction. As a rule, the Abstract have a limited space and should described in short form the main results.
The Abstract like: Now is very popular study of… for me is unacceptable. However, if editor recommended, I will be forced to rewrite
- The introduction must be revised. Add more state-of-the-art comparisons for the proposed work. Then do a critical analysis of previous research. State explicitly the shortcomings of previous research. What is positive in previous research and what is negative. Based on that, you explicitly define the goal of the research and the scientific hypothesis.
We given in Introduction the references on two new books and five recent reviews in this field. The goal and novelty of study are postulated in the last paragraph.
- Highlight the novelty of your methodology.
The novelty of our methodology connected with chemistry of block-copolymer synthesis. The characterization is traditional as in most of chemical papers.
- The biggest shortcoming of the research is that there is a poor presentation of Figures. Add intensity on the Y-axis to understand the observations.
The intensity on the Y-axis of NMR spectra is useless because the amount of resonance nucleus is proportional of signal square for each nucleus. As a result, the Y-axis at presentation of NMR spectra never quantified. Other situation is at presentation of IR spectra where Y-axis quantified in relative absorption units. However, if it is presented several IR spectra this calibration make useless because the absorption depended not only on chromophore properties but also from samples thickness which is impossible precise controlling for different samples if studying films.
- Add more details about the Adhesion strength testing.
Adhesion of copolymers to metals was tested according to ISO 4624:2002. As all ISOs in this one is given detail description of testing procedure.
- Highlight the observation in SEM images by arrows and circles.
The SEM images did not contain any special peculiarities located in fixed regions.
- How did you choose the experiment setting? Elaborate experiments parameters.
We based on ISO recommendations and good lab practice.
- The Conclusion section should be rewritten. Highlight your scientific contribution. Highlight the benefits of your research. Define shortcomings and future research.
See point 1
